**DOI: 10.1038/ncomms14315**　　**OPEN**

# Rapid evolution of dispersal ability makes biological invasions faster and more variable

Brad M. Ochocki[1] & Tom E.X. Miller[1]

Genetic variation in dispersal ability may result in the spatial sorting of alleles during range expansion. Recent theory suggests that spatial sorting can favour the rapid evolution of life history traits at expanding fronts, and therefore modify the ecological dynamics of range expansion. Here we test this prediction by disrupting spatial sorting in replicated invasions of the bean beetle *Callosobruchus maculatus* across homogeneous experimental landscapes. We show that spatial sorting promotes rapid evolution of dispersal distance, which increases the speed and variability of replicated invasions: after 10 generations of range expansion, invasions subject to spatial sorting spread 8.9% farther and exhibit 41-fold more variable spread dynamics relative to invasions in which spatial sorting is suppressed. Correspondingly, descendants from spatially evolving invasions exhibit greater mean and variance in dispersal distance. Our results reveal an important role for rapid evolution during invasion, even in the absence of environmental filters, and argue for evolutionarily informed forecasts of invasive spread by exotic species or climate change migration by native species.

[1] Department of BioSciences, Program in Ecology and Evolutionary Biology, Rice University, 6100 Main Street, MS-170, Houston, Texas 77005-1892, USA. Correspondence and requests for materials should be addressed to B.M.O. (email: brad.ochocki@rice.edu).

Understanding and predicting the dynamics of biological invasions are among the leading environmental challenges in the Anthropocene. Biological invasions play out in the contexts of intentionally or accidentally introduced species[1], recovery of threatened species[2] and, increasingly, shifts in distributional limits in response to climate change[3]. Long-standing ecological theory provides a framework for understanding and predicting the velocity of spread based on life history traits related to dispersal ability and reproductive potential[4]. However, accurate invasion forecasts remain elusive[5–8], at least partly because of the substantial variability in spreading speed that appears to be an intrinsic but poorly understood feature of biological invasions[9–11].

Recent theory suggests that evolutionary processes unique to spreading populations can influence the ecological dynamics of invasion by modifying traits related to dispersal, reproduction or both[12–15]. The dispersal phase of invasion provides an opportunity for individuals to spatially sort themselves. Spatial sorting is expected to cause the over-representation of highly dispersive phenotypes at the leading edge of an invasion wave, increasing the probability of assortative mating between highly dispersive individuals and, if dispersal is heritable, the probability that offspring produced at the leading edge will also be highly dispersive[14]. Because the leading edge is characterized by low population density, highly dispersive individuals may leave more descendants, per capita, because of greater resource availability and reduced intraspecific competition. This combination of spatial allele sorting and increased per capita growth at the leading edge of the invasion wave—a process described in the invasion literature as 'spatial selection'—is predicted to favour the evolution of increased dispersal at the invasion front[13–17]. Finally, in addition to spatial selection for increased dispersal, the low-density leading edge is also subject to natural selection, which may favour the evolution of increased reproductive rate in the absence of strong resource limitation ($r$-selection)[13]. Since invasion speed is determined both by dispersal and low-density reproductive rate, these evolutionary processes that occur as a result of spatial allele sorting are expected to accelerate range expansion[15].

Despite well-developed theory, empirical evidence that the ecological dynamics of invasion can be modified by the spatial sorting of alleles is scarce[16–18]. Most relevant studies have been observational and retrospective, comparing demography and dispersal trait values between range-core and range-edge populations[15,19–23]. These studies highlight potential for rapid trait evolution during range expansion but reveal little about how trait evolution modifies the ecological dynamics of invasion. Several considerations raise uncertainty about the general ecological importance of spatial evolutionary mechanisms. First, theoretical models of spatial selection usually assume perfect heritability of dispersal behaviour[12,24,25]. In reality dispersal is a complex, likely polygenic trait[26] with imperfect heritability[20,26–29]. Since low trait heritabilities generally dampen evolutionary change[30], imperfect dispersal heritability may cause responses to spatial selection to be weaker than predicted by theory. Second, stochastic forces may overwhelm the directional influence of spatial selection at the expanding edge: low-density, leading edge patches—precisely the locations where spatial selection on dispersal and reproduction is expected to be most potent—are also the locations most affected by stochasticity in demography[10,31], dispersal[10] and allelic composition because of founder events[32]. Finally, even in systems where spatial selection has been implicated in accelerating spread[15], the influence of alternative, non-evolutionary accelerating mechanisms[4,33] is difficult to exclude. Experimental tests are thus critical for understanding whether and to what extent the evolutionary processes that arise from spatial allele sorting modify spread dynamics—and therefore how much information is lost by ignoring them, as most ecological studies currently do.

Laboratory-based mesocosms provide a powerful setting to study spread dynamics because they distil invasions to their essential ingredients—local population growth and dispersal—and facilitate experimental manipulations that are impractical in the field. Importantly, replicated mesocosms reveal a distribution of invasion trajectories in simple settings, providing a window into intrinsic sources of variability that cannot be achieved with unreplicated invasions across heterogeneous natural landscapes[9,10,16,17,34]. We used replicated laboratory invasions of the bean beetle *Callosobruchus maculatus* to test the influence of spatial evolutionary processes on the ecological dynamics of range expansion. Bean beetles provide a powerful and convenient system for testing rapid evolution during invasions because they are semelparous with fast (30 days) and nonoverlapping generations. Our experimental set-up allows beetle populations to spread through homogenous, one-dimensional (1D) landscapes of interconnected habitat patches, with opportunity for dispersal following within-patch population growth (Methods), giving rise to travelling invasion waves over successive generations.

We experimentally prevented spatial allele sorting (and thus the spatial evolutionary processes that act on dispersal and reproduction) in a subset of replicate invasions ($N = 9$) by shuffling the post-dispersal locations of all individuals: we mixed all dispersed beetles into a global pool and then randomly redistributed them back on the landscape, maintaining the density and sex ratio of each patch exactly as we found them. This method decouples an individual's genotype from its spatial position, preventing highly dispersive or fecund individuals from carrying their alleles to the invasion fronts. We compared spread dynamics of these shuffled invasions to those of spatially sorted (control) invasions ($N = 9$). After 10 generations, we reared beetles from the leading edge of each invasion replicate in a 'common garden' setting to measure whether spatial evolutionary processes generated genetically based differences in dispersal distance or low-density reproductive rate between treatments.

Our experiments show that spatial sorting increases the speed and variability of replicated invasions. These changes are caused by the rapid evolution of dispersal: we observe that descendants from spatially sorted invasions exhibit greater mean and variance in dispersal distance compared with descendants from spatially shuffled invasions. These results reveal an important role for rapid evolution during invasion, even in the absence of environmental filters, and argue for evolutionarily informed forecasts of biological invasions.

## Results

**Effects of spatial sorting on invasion dynamics**. Spatial evolutionary processes increased the speed of range expansion (Figs 1 and 2, Supplementary Table 1). After 10 generations, spatially sorted invasions spread 8.9% farther, on average, than shuffled invasions (Figs 1 and 2). Spatial sorting also increased variability across realizations of the invasion process, independently of the increase in the mean invasion speed (Fig. 2 and Supplementary Figs 1 and 2). The greater variability of spatially sorted invasions was attributable to among- and not within-replicate variance (Supplementary Fig. 1; 100% combined widely applicable information criterion (WAIC) support for models including a treatment effect on among-replicate variance, Supplementary Table 2). The contrast between treatments was stark: among-replicate variance in speed was 41.4 times larger in the sorted replicates versus the shuffled replicates, a fold-change that far exceeded that

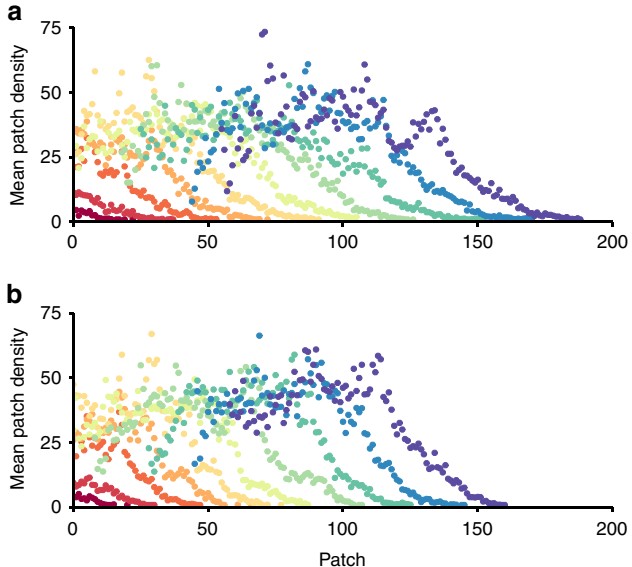

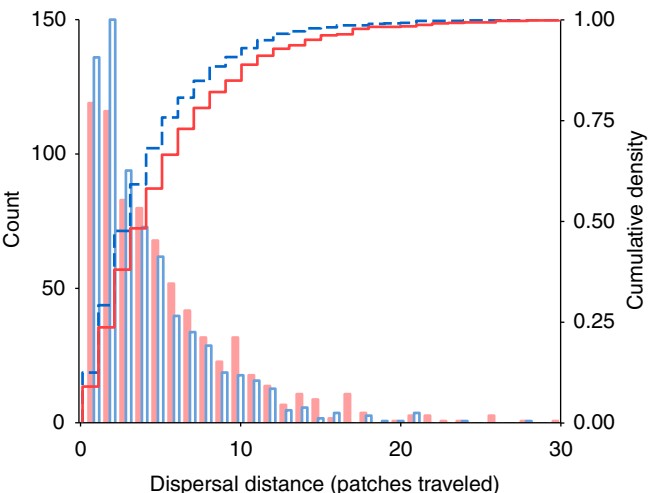

**Figure 1 | Wave-like expansion of experimental beetle invasions.** Points show the mean population density of each patch, averaged across replicates, for (**a**) spatially sorted and (**b**) spatially shuffled invasions (nine replicates each). Invasions spread from left to right across the *x* axis (patch number) over time so that each coloured wave represents a different invasion generation. On average, spatially sorted replicates travelled farther and faster than shuffled replicates after 10 generations.

**Figure 3 | Comparison of post-invasion dispersal kernels.** Histograms (left axis) show dispersal distances of beetles descended from spatially sorted (filled red bars) and shuffled (open blue bars) invasions following one generation in a common environment; results from a second generation in a common environment were qualitatively similar (Supplementary Figs 4 and 5). Lines show corresponding empirical cumulative distribution functions (right axis; sorted: red solid line; shuffled: blue dashed line). Beetles from spatially sorted invasions dispersed farther on average than beetles from shuffled invasions ($N = 3,240$, $D^+ = 0.0646$, $P \ll 0.0001$). A hierarchical Bayesian analysis that accounted for random variance across replicates and fitted male and female kernels separately yielded similar results (Methods, Supplementary Tables 4 and 5).

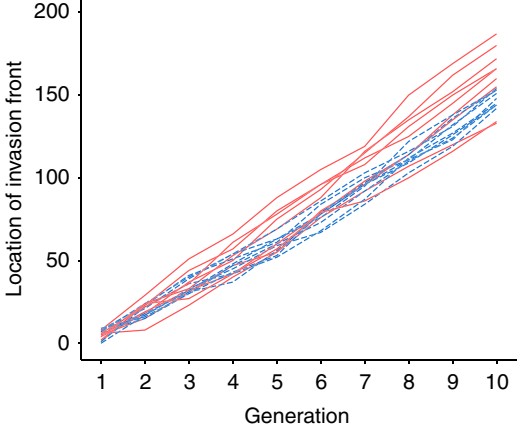

**Figure 2 | Spread dynamics of spatially sorted and shuffled invasion replicates.** Lines show raw invasion trajectories (invasion extent through time) for all replicates in the spatially sorted (red solid lines) and shuffled (blue dashed lines) invasions (nine replicates each). Spatially sorted replicates invaded faster, on average, than shuffled replicates (model selection results in Supplementary Table 2). Variance in spread from one generation to the next (within-replicate variance) was similar between treatments but spatially sorted replicates had a larger among-replicate variance in invasion speed (Supplementary Figs 1 and 2 and Supplementary Table 3).

of the mean invasion speed. Thus, the elevated variance was due to unique trajectories of individual invasion replicates and not inconsistency in speed from one generation to the next. In fact, while most spatially sorted replicates were faster than most shuffled replicates, the two slowest replicates were spatially sorted (Fig. 2).

**Evolved trait differences between treatments.** A pre-invasion analysis confirmed that *C. maculatus* is capable of evolution via

spatial selection because of the existence of additive genetic variance for dispersal distance (Methods, Supplementary Table 3 and Supplementary Fig. 3). In post-invasion common-garden experiments, descendants from spatially sorted invasion fronts dispersed farther, on average, than descendants from shuffled invasion fronts (males and females combined, Kolmogorov–Smirnov test: $D^+ = 0.0646$, $P = 1.4E - 6$; Fig. 3) and also exhibited greater among-replicate variance in dispersal; the magnitudes of these effects differed by sex. Females descended from sorted invasions had dispersal kernels with greater means, similar tails and similar among-replicate variance in kernel parameters when compared with females descended from shuffled invasions (90% of cumulative WAIC weight for an effect of treatment; Supplementary Table 4 and Supplementary Fig. 4). Males descended from spatially sorted invasions had dispersal kernels with similar means, longer tails and higher among-replicate variance in kernel parameters when compared with males descended from shuffled invasions (97% of cumulative WAIC weight for an effect of treatment; Supplementary Table 5 and Supplementary Fig. 5). For both males and females, common garden generation had an effect on dispersal kernel shape independent of treatment (Supplementary Tables 4 and 5). In the second common garden generation, beetles descended from all invasions dispersed less far and had shorter-tailed dispersal kernels than in the first generation. However, the differences between treatments were qualitatively similar between generations, with greater evolved dispersal ability in the spatially sorted treatment (Supplementary Tables 4 and 5; Supplementary Figs 4 and 5).

Estimates of pre-invasion dispersal kernels indicate that there were no initial differences in dispersal ability between treatments (Supplementary Fig. 6). For female beetles, model selection indicated that a null model lacking any treatment differences provided the best fit to dispersal data from the first generation of spread (47% WAIC support; Supplementary Fig. 6a and

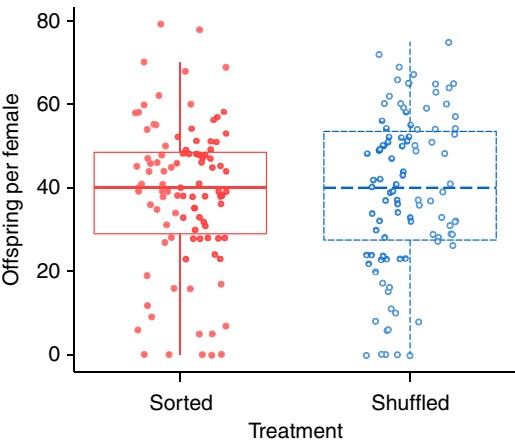

**Figure 4 | Comparison of post-invasion reproductive rates.** Tukey boxplots showing the number of offspring per female in spatially sorted (red solid boxes, closed circles) and shuffled (blue dashed boxes, open circles) replicates, following one generation in a common environment (nine replicate populations per treatment, 11 females per replicate, $N = 198$). Results from a second generation in a common environment were similar (Supplementary Fig. 7). Model selection using generalized linear mixed models (GLMMs) suggests no difference between treatments (model selection results in Supplementary Table 7). Points show raw data, and are jittered along the x axis to reduce overlap.

Supplementary Table 6). For males, model rankings were closer but a model including treatment differences was not considerably better than the null model (38% WAIC support for the model containing an effect of treatment on variance, 35% WAIC support for the null model; Supplementary Fig. 6b and Supplementary Table 6). These results indicate that post-invasion differences in dispersal between treatments evolved during range expansion.

In contrast to dispersal, the post-invasion common gardens showed no difference in low-density reproductive rate between treatments (52% Akaike information criterion (AIC) weight for the null model; Fig. 4, Supplementary Fig. 7 and Supplementary Table 7). The number of offspring produced by females in both treatments was also similar across common garden generations.

## Discussion

Our results provide novel experimental evidence that the spatial sorting of alleles can, on ecological timescales, generate accelerated invasions via the rapid evolution of dispersal ability. Most studies of rapid evolution during biological invasion have focused on adaptations to novel environmental selective pressures and their potential to promote invasiveness[35–37]. Importantly, we find an influence of evolution on invasion dynamics even in homogeneous landscapes, absent environmental filters, because space *per se* is an evolutionary agent. These experiments are among the first to validate a wealth of theoretical research on the eco-evolutionary dynamics of range expansion[11,12,14,15,17,20,38], but also paint a more complex picture than is currently appreciated. Not only did we detect a signature of evolutionary acceleration despite intrinsic variability in spread, we also found that evolution was a source of that variability. In fact, the evolutionary effect on variability in spread far exceeded the effect on the mean.

Evolved differences in dispersal kernels of sorted and shuffled invasions were consistent with their contrasting spread dynamics: descendants from spatially sorted invasions exhibited kernels with greater means (in females) and longer tails (in males) compared with descendants from shuffled invasions, which would increase the probability of long-distance dispersal events. Since

we observed no differences in pre-invasion dispersal kernels, the treatment-specific differences in post-invasion kernels reflect evolutionary divergence that developed during range expansion. Furthermore, the post-invasion common gardens showed no difference in low-density reproductive rate between treatments. This leads us to conclude that, while evolution of the reproductive rate may accompany evolution of dispersal during range expansion[13,15], the faster and more variable spread of spatially sorted invasions that we observed was due primarily to rapid evolution of dispersal ability with little contribution of the reproductive rate. Given the relatively short timescale and modest population sizes of our experiment, evolutionary changes in dispersal were likely caused by the redistribution of standing allelic variation rather than novel mutations.

We hypothesize that 'gene surfing' is a likely mechanism[24,32,39–42] by which spatially sorted invasions exhibited amplified variance in spread dynamics[11]. Gene surfing is a drift-like process where alleles initially present at the leading edge of an incipient invasion can become fixed because of the serial founder events that are characteristic of an expanding invasion wave. Much like genetic drift, gene surfing can cause allele fixation at expanding fronts even in homogenous, non-selective environments[24,32,39–41] and even if the allele is deleterious[41,42]. The serial genetic bottlenecks that promote gene surfing (that is, rare, leading edge colonists are likely the offspring of leading edge colonists from the previous generation) were likely to be especially influential in our experimental invasions, where vanguard patches were often colonized by one or two individuals. Although the densities of leading edge patches were similar between our treatments (Supplementary Table 8), gene surfing would have been suppressed by the shuffle treatment along with the directional influence of spatial selection, since shuffling brings a new, random set of alleles to the invasion front in each generation.

The stochastic nature of gene surfing means that each realization of range expansion can propagate a unique allelic composition even if initiated from a common source pool, as in our experiments[43]. Indeed, our common garden results reveal not only increased dispersal distance, on average, in spatially sorted invasions but also elevated among-replicate variance in dispersal (in males). Furthermore, comparison of pre- and post-invasion dispersal kernels shows that some spatially sorted replicates show strong evolved increases in dispersal while other replicates show weaker evolved decreases in dispersal (Supplementary Fig. 8), consistent with increased mean and among-replicate variance. We hypothesize that the net effects of spatial evolutionary processes reflect a balance of directional spatial selection and stochastic gene surfing; these forces resulted in dispersal kernels, and consequently invasion speeds, that were not only greater, on average, but also more variable.

Our results accompany a recent burgeoning of experimental tests of theory for spatial sorting and its influence on invasion dynamics[16–18]. Collectively, these studies present some coherent patterns of evolution during range expansion, particularly the result that spatial sorting can generate evolutionarily accelerated invasions, even in homogenous environments. However, disparities between these studies make it clear that invasion context—the stage on which these evolutionary processes play out—is critically important for determining the outcome of evolution and its ecological consequences. While two new studies (including ours) with sexually reproducing organisms indicate that evolution drives increased variance in invasion dynamics[18], other recent work with asexually reproducing plants found the opposite: evolution reduced variance across invasion replicates[17]. The difference in breeding systems may explain this disparity, since the variance-promoting effects of gene surfing are likely less

prominent in populations without genetic recombination. Instead, for invasions consisting of mixtures of asexual lineages, the 'fastest' lineage would likely dominate the expanding fronts, such that the directional and homogenizing influence of spatial selection is the dominant evolutionary force. Furthermore, while theory suggests that spatial evolutionary processes should result in the enhancement of 'r-selected' traits, such as increased fecundity or faster life histories[13], evidence from recent experiments and retrospective studies is mixed. Specifically, studies where spatial sorting is thought to promote dispersal ability have shown fecundity to be reduced[18,23] or unchanged (our study), although at least one study found evidence for the evolution of a faster life cycle[22]. One possible explanation for these mixed results is genetic constraints, such as correlations between dispersal and other life history traits, which are currently unaccounted for in most studies; these may have important effects on evolutionary outcomes[25,44–46], a hypothesis that merits further study.

Lastly, it has been suggested that there may be fundamental limits on our ability to predict invasive spread because accounting for ecological sources of stochasticity fails to capture the total observed variability[9]. Our results suggest that evolutionary sources of stochasticity may be the missing piece of this puzzle and offer a more optimistic assessment of predictive capacity. While accurate point estimates for the pace of invasion will likely remain elusive, accounting for evolutionary processes in ecological forecasts can yield improved uncertainty windows for the distribution of possible invasion outcomes. With more accurate bounds on predicted invasion outcomes, natural resource managers may be better equipped to prepare for best- and worst-case scenarios of expansion by native or exotic species.

## Methods

**Establishing the source population.** Laboratory populations, such as those of *C. maculatus* used here, are typically highly inbred and the potential loss of genetic variation due to extensive inbreeding may be unrepresentative of natural populations. We minimized the influence of historical inbreeding on our experiment by randomly selecting females ($N = 20$) and males ($N = 20$) from each of 10 inbred lineages, originally isolated from different parts of the species' global distribution[47], to create a well-mixed and genetically diverse founding population of 400 beetles with a 1:1 sex ratio. Beetles from different lineages are known to readily interbreed and create viable offspring[48,49]. We provided virtually unlimited resources ($\sim 400$ g black-eyed peas; *Vigna unguiculata unguiculata*, Fabaceae) for 10 generations to allow for population growth, genetic admixture and to reduce linkage disequilibrium[50]. Beetles were maintained in a climate-controlled growth chamber on a 16:8 photoperiod at 28 °C.

**Invasion experiments.** We discretized the beetle life cycle into a 30-day local demography phase (mating, oviposition, development and eclosion) and a 2-h dispersal phase. We initiated replicate invasions in the dispersal phase by randomly selecting 25 males and 25 females from the source population and introducing them into a 1D landscape consisting of 144 patches (Petri dishes), each containing seven black-eyed peas, with dispersal connections (1/4″ length of 1/8″ tubing) between adjacent patches. The 50 founding beetles were introduced at equal initial densities to the first five patches (five males and five females per patch). After 2 h of dispersal, the connecting tubes between patches were blocked with pipe cleaners, and the numbers of females and males in each patch were counted. For each invasion replicate assigned to the shuffle treatment, we then mixed all beetles into a 'global' pool and randomly redistributed them back on the landscape, maintaining the density and sex ratio of each patch exactly as we found them. For spatially sorted (control) replicates, beetles were allowed to mate and oviposit in the patches to which they dispersed. Since replicates were manipulated differently depending on treatment, experimenters were not blind to treatment. Post-dispersal, all beetles in each patch were transferred to unconnected Petri dishes, each containing seven black-eyed peas ($\sim 1.4$ g) for the demography phase of the life cycle. Limiting the number of black-eyed peas to seven imposed resource limitation, resulting in negative density-dependent population growth and a carrying capacities of roughly 40 beetles (Fig. 1). Therefore, in addition to spatial selection on dispersal ability, we hypothesized that beetles at low-density invasion fronts may be under natural selection for increased fecundity.

After 30 days, offspring were transferred back to their 'home' location in the landscape (the patch where their mother laid them), beginning the dispersal phase of the next generation. This sequence of dispersal, counting/sexing, shuffling (if

applicable) and local demography was repeated for 10 generations. To keep the experiment logistically manageable, we removed patches from the trailing edges of the invasions; we applied the rule that only offspring from the leading 60 patches of each replicate were allowed to participate in the next generation[10,34]. For most replicates, we began removing trailing patches after the fifth generation. We also discarded any beetles that dispersed more than two patches to the left of patch 0, and only tracked the right-spreading invasion wave[10,34].

Owing to the large number of beetles in each replicate, it became unfeasible to sex beetles in the invasion core after the sixth generation. Consequently, for generations seven through ten, we only sexed beetles on the expanding front of the invasion wave, using the following method. Starting at the farthest occupied patch and moving towards the range core, we sexed beetles in every patch until there were a total of three patches that contained 40 or more beetles; after this patch, we continued counting beetles, but no longer identified their sex. We chose a density of 40 beetles because that roughly represents the local carrying capacity, given the seven-bean resource environment (Fig. 1), and we chose three patches to ensure that we were safely into the core of the invasion before disregarding sex. In deep-core patches where sex ratios were not measured, we returned the required numbers of beetles from the global pool irrespective of sex. Because sex ratios are roughly constant in the invasion core[10], this methodology preserved the average, unmeasured sex ratio.

For each replicate, we measured invasion extent in each generation as the distance from the starting patch (patch 0) to the farthest patch that contained at least four beetles. We chose a threshold density of four beetles because it limited the influence of observation error (e.g., failing to detect patches that contained single beetles). We do not assume that the leading four-beetle patch contains a viable population; the density threshold simply represents a marker with which to measure displacement of the invasion from one generation to the next[10,34]. Additional analyses showed that our results were not sensitive to the threshold of four beetles; since we are attempting to measure a travelling wave, any density that consistently corresponds to a location on the wave's leading edge should yield similar results[10,34].

To evaluate the influence of the shuffle treatment on spread dynamics, we performed model selection on the spatial spread data in two steps. First, we sought to determine the model that best described how treatment affected mean invasion extent over time. We used a first-order autoregressive (AR(1)) linear mixed model to account for the fact that invasion extents within replicates were spatiotemporally autocorrelated. The invasion extent of replicate $i$ in treatment $j$ and generation $k$ was modelled according to:

$$E_{ijk} \sim \text{Normal}\left(\mu_{ijk}, \sigma_W^2\right) \tag{1}$$

$$\mu_{ijk} = \beta_0 + \beta_{\text{TRT}}\text{TRT}_j + (\beta_{\text{GEN}} + \gamma_i)\text{GEN}_k + \beta_{\text{TRT}\times\text{GEN}}\text{TRT}_j\text{GEN}_k + \beta_{\text{AR}}E_{ij(k-1)} \tag{2}$$

$$\gamma_i \sim \text{Normal}\left(0, \sigma_A^2\right) \tag{3}$$

where $\mu_{ijk}$ is the expected invasion extent and $\sigma_W^2$ is residual variance by which each replicate deviates from its expected value (hereafter 'within-replicate variance'). $\beta_0$ is the model's intercept, $\beta_{\text{TRT}}$ represents the effect of the shuffle treatment, $\beta_{\text{GEN}}$ represents the slope of the invasion extent with respect to generation (that is, invasion velocity), $\beta_{\text{TRT}\times\text{GEN}}$ allows the slope of invasion extent over time to vary between treatments and $\beta_{\text{AR}}$ is the autoregressive term that accounts for the invasion extent in the previous generation. A replicate-specific random effect in invasion velocity, $\gamma_i$, is centred at the mean velocity $\beta_{\text{GEN}}$ and has among-replicate variance $\sigma_A^2$. We also tested for nonlinearly increasing invasion speeds by testing models that accounted for the square of generation (Supplementary Table 1). To identify the best model for the mean invasion extent, we fit all possible combinations of fixed effects in candidate models using the 'nlme' package[51] in R, and used AIC to select the best-fitting model (Supplementary Table 1). Visual inspection of the residuals confirmed that the model specification was appropriate for our data set.

Once we determined the best-fit model for mean invasion extent, we tested whether including treatment-specific within- and/or among-replicate variances improved fit, by allowing $\sigma_W^2$ and $\sigma_A^2$ to vary between treatments. We considered models where both, either or neither variance term was treatment-dependent, yielding four models in total, using the fixed-effect linear predictors identified by the model selection above. The model formulation was similar to equations 1–3 but expanded to include the possibility of treatment-specific variances. The full model was:

$$E_{ijk} \sim \text{Normal}\left(\mu_{ijk}, \sigma_{W_j}^2\right) \tag{4}$$

$$\mu_{ijk} = \beta_0 + \beta_{\text{TRT}}\text{TRT}_j + \left(\beta_{\text{GEN}} + \gamma_{ij}\right)\text{GEN}_k + \beta_{\text{TRT}\times\text{GEN}}\text{TRT}_j\text{GEN}_k + \beta_{\text{AR}}E_{ij(k-1)} \tag{5}$$

$$\gamma_{ij} \sim \text{Normal}\left(0, \sigma_{A_j}^2\right) \tag{6}$$

We fit candidate models using Stan[52] in R (ref. 53) and calculated the WAIC, using the 'loo' package[54], to determine the best-fitting model[55] (Supplementary Table 2).

As an additional test to determine whether spatial selection generated increased variance in invasion extent over time, we calculated the coefficient of variation (CV) within each treatment, across generations. In addition to the linear mixed model analysis above, calculating the CV is another method to estimate whether the increase in variance that we observed was independent of the increase of the mean (Supplementary Fig. 2).

**Common garden experiment.** We used a 'common garden' approach to test for trait evolution after 10 generations of range expansion. After the dispersal phase of the 10th generation, we removed the farthest 10 male and 10 female beetles from each replicate and placed them into common garden environments with 200 g black-eyed-peas—virtually an unlimited amount of resources. We did not perform the shuffle manipulation on the 10th generation of invasion, so the beetles that we sampled arrived at the invasion fronts by their own means. Each replicate-specific common garden was placed in an incubator under standard conditions (16:8 photoperiod at 28 °C), where beetles were able to mate and reproduce. After 30 days of development, we measured dispersal and lifetime-reproductive-success for a random sample of the offspring that emerged from the common garden. In addition to measuring traits in these offspring, we randomly selected 10 males and 10 females from each replicate to create a second generation of replicate-specific common garden populations, reared under identical conditions, and measured traits in the offspring from those populations.

Dispersal trials took place in the same 1D, homogenous environment in which we conducted the invasion experiments. Resources were evenly distributed among all patches (seven black-eyed peas per patch). Since dispersal might be affected by local demographic conditions, we held the number of dispersers and their sex-ratio constant during dispersal trials; we performed 3 dispersal trials for each replicate, using 15 males and 15 females in each trial for a total of 45 dispersal observations per sex per replicate. For each trial, beetles were placed in a common starting patch and allowed to disperse for two hours, as in the invasion experiment. During the dispersal period, arrays were placed in incubators under our standard conditions. After two hours, the net displacement (number of patches) of each male and female was recorded. We performed the same dispersal measurements for both common garden generations.

To estimate fertility, we randomly sampled 11 females from each common garden replicate, and placed each female in a Petri dish containing 25 g black-eyed peas and a single male. Females were allowed to mate and oviposit until they died. After 30 days of incubation, we counted all fully-eclosed adult beetles as the lifetime-reproductive-success for each female. We repeated the fertility measurements for two common garden generations.

We tested for evolved differences in dispersal distances between common garden populations of control and shuffle replicates in two ways. First, we compared the empirical cumulative distribution functions (ecdf) for both treatments, combining all data within treatments, using a discrete Kolmogorov–Smirnov test[56,57] (Fig. 3). This analysis provides a simple test of treatment effects without needing to specify the probability distribution of dispersal distance. However, this test does not include effects of sex or common garden generation and it disregards the non-independence of multiple observations within invasion replicates.

For a more thorough follow-up analysis, we modelled dispersal distance as a random draw from a discrete probability distribution (that is, a dispersal kernel). In preliminary analyses that compared the fits of different discrete distributions to the dispersal data, we found that a Poisson-inverse Gaussian (PIG) distribution[58,59], which allows for a wider range of kurtosis than the Poisson or negative binomial distributions, provided the best fit. The PIG is a two-parameter distribution described by a mean $\xi$ and shape parameter $\omega$, with variance given by $\xi(1 + \xi/\omega)$; it is a special case of the Sichel distribution[58,59] with $\gamma = -0.5$. We analysed female and male dispersal kernels separately since they are known to differ[10] and may have responded differently to spatial selection. The full model for dispersal distance ($d$) of individual $i$ from replicate $j$, treatment $k$, generation $l$ was:

$$d_{ijkl} \sim \mathrm{PIG}\left(\xi_{jkl}, \omega_{jkl}\right) \quad (7)$$

$$\log(\xi_{jkl}) = \alpha_0 + \alpha_{\mathrm{TRT}}\mathrm{TRT}_k + \alpha_{\mathrm{CGG}}\mathrm{CGG}_l + \alpha_{\mathrm{TRT}\times\mathrm{CGG}}\mathrm{TRT}_k\mathrm{CGG}_l + \varepsilon_{jk} \quad (8)$$

$$\log(\omega_{jkl}) = \beta_0 + \beta_{\mathrm{TRT}}\mathrm{TRT}_k + \beta_{\mathrm{CGG}}\mathrm{CGG}_l + \beta_{\mathrm{TRT}\times\mathrm{CGG}}\mathrm{TRT}_k\mathrm{CGG}_l + \gamma_{jk} \quad (9)$$

$$\varepsilon_{jk} \sim \mathrm{Norm}\left(0, \sigma_{\xi_k}^2\right) \quad (10)$$

$$\gamma_{jk} \sim \mathrm{Norm}\left(0, \sigma_{\omega_k}^2\right) \quad (11)$$

where $\alpha_0$ is the intercept for mean $\xi_{jkl}$, $\alpha_{\mathrm{TRT}}$ is the effect of treatment $k$, $\alpha_{\mathrm{CGG}}$ is the effect of common garden generation $l$, and $\alpha_{\mathrm{TRT}\times\mathrm{CGG}}$ represents the interaction between treatment and common garden generation. The $\beta$ parameters in equation 9 that model the shape parameter $\omega_{jkl}$ have identical notation. The random effect terms for replicate $j$ treatment $k$ are $\varepsilon_{jk}$ and $\gamma_{jk}$, and these are normally distributed with treatment-specific variances $\sigma_{\xi_k}^2$ and $\sigma_{\omega_k}^2$, respectively. We compared nested versions of the above full model (Supplementary Tables 4 and 5), considering effects of treatment and generation on neither or both kernel parameters. In addition to models with treatment-specific variances in random

effects (e.g., $\sigma_{\omega_k}^2$), we also considered models with no effect of treatment on the replicate variances. We fit all combinations of fixed and random effects using Stan[52] in R (ref. 53), and calculated WAIC using the 'loo' package[54] to select the best-fitting model (Supplementary Tables 4 and 5; Supplementary Figs 4 and 5).

We tested for differences in beetle fertility by fitting the total number of adult offspring from each female to a linear model with a negative binomially-distributed response, and tested for effects of treatment and common garden generation. The total number of offspring from female $i$ in replicate $j$, treatment $k$, generation $l$ was:

$$\mathrm{offspring}_{ijkl} \sim \mathrm{NegBinom}\left(\mu_{jkl}, \phi\right) \quad (12)$$

$$\log\left(\mu_{jkl}\right) = \beta_0 + \beta_{\mathrm{TRT}}\mathrm{TRT}_k + \beta_{\mathrm{CGG}}\mathrm{CGG}_l + \beta_{\mathrm{TRT}\times\mathrm{CGG}}\mathrm{TRT}_k\mathrm{CGG}_l + \varepsilon_j \quad (13)$$

$$\varepsilon_j \sim \mathrm{Norm}\left(0, \sigma^2\right) \quad (14)$$

We allowed fixed and random effects to modify mean fertility; the overdispersion parameter $\phi$ was fit as a constant. Replicate-specific random effects ($\varepsilon_j$) were modelled as being normally distributed about 0. We fit all nested versions of the above model using the glmmADMB package[60] in R and used AIC to select the best-fitting model (Fig. 4, Supplementary Fig. 7 and Supplementary Table 7).

**Testing for additive genetic variance in dispersal distance.** After our source population had gone through six generations of genetic admixture but before the invasion experiment, we conducted a nested paternal half-sib breeding experiment[30,61] to test whether there was additive genetic variance for dispersal ability. The breeding design requires that each male (sire) be mated with multiple females (dams); the result is a nested pedigree, where all full siblings (offspring from the same sire and dam) are a subset of half-siblings (offspring from the same sire but different dams). Using this nested design, it is possible to estimate additive genetic variance by estimating the variance among half-sibling families (that is, the sire variance), which is proportional to the additive genetic variance[30]. We isolated virgin adult beetles ($N = 17$ males, 51 females) into breeding groups containing one male and three females ($N = 17$ groups yielding 51 full-sib families). After 48 h of mating, each female was transferred to a Petri dish containing $\sim 9.5$ g black-eyed peas and permitted to oviposit for 24 h.

Offspring began to eclose after $\sim 29$ days of incubation and dispersal trials began within 48 h of emergence. Since dispersal might be affected by local demographic conditions, we held the number of dispersers, their sex ratio and relatedness constant during dispersal trials: trials included five females and five males from full-sibling groups. Occasionally, it was not possible to disperse five females and five males from the same full-sibling group because of asynchronous maturation, in which case we supplemented trials with marked males and females from the source population to reach the requisite numbers; however, all trials contained at least three female and two male full siblings, and only the data from full siblings were used in analysis. We ran dispersal trials opportunistically, as beetles emerged, in an effort to collect as much data as possible. The total number of offspring that emerged among full-sib families varied, with some families yielding no offspring at all. This resulted in unequal numbers of replicate dispersal trials among full-sib families (min = 0, max = 9, median = 2), but our statistical approach is robust to these differences.

For each trial, beetles were placed in a common starting patch and allowed to disperse for 2 h, as in the invasion experiment. During dispersal, arrays were placed in lighted growth chambers (28 °C). After 2 h, the net displacement (number of patches) of each male and female from the starting patch was recorded and used to estimate sex-specific dispersal kernel estimates for each full-sib family.

We used a hierarchical model selection approach to determine whether or not accounting for sire variance significantly improved the fit of dispersal kernels to the data[30,61]. We modelled net dispersal distance ($d$) of sibling $i$ from dam $j$ and sire $k$ as being drawn from the PIG distribution[58,59], such that:

$$d_{ijk} \sim \mathrm{PIG}\left(\xi_{jk}, \omega\right) \quad (15)$$

$$\log\left(\xi_{jk}\right) = \alpha_0 + \varepsilon_{jk} \quad (16)$$

$$\log(\omega) = \beta_0 \quad (17)$$

where $\alpha_0$ is the intercept for mean $\xi_{jk}$, $\beta_0$ is the intercept for shape parameter $\omega$ and $\varepsilon_{jk}$ is a random effect that allows for family-specific variation in kernel mean parameter values. We did not model family-specific random effects in dispersion parameter $\omega$ because some families had very few dispersers (minimum number of dispersers = 4, maximum = 64, median = 27), which made it difficult to estimate family-specific dispersion and caused model convergence failures. However, we achieved reliable convergence when modelling random effects in $\xi_{jk}$ only. Finally, since previous studies have detected sex differences in dispersal distance for *C. maculatus*[10], we analysed male and female dispersal kernels separately.

We considered two competing models in which sire variance was either included or not; these two models differed only in the way random effects were modelled. When $\varepsilon_{jk}$ accounted for sire variance (and thus additive genetic variance), its error was modelled as a nested random effect, with the random effect

                                                              

of dam $j$ nested within the random effect of sire $k$ ($S_k$), such that:

$$\varepsilon_{jk} \sim \text{Norm}\left(S_k, \sigma_D^2\right) \tag{18}$$

$$S_k \sim \text{Norm}\left(0, \sigma_S^2\right) \tag{19}$$

The variance term $\sigma_D^2$ represents the among-dam variance (maternal effects), and the variance term $\sigma_S^2$ represents the variance among sires and is proportional to the additive genetic variance[50].

For the alternative model without additive genetic variance in the dispersal kernel, family-specific random effects were modelled as:

$$\varepsilon_{jk} \sim \text{Norm}\left(0, \sigma_D^2\right) \tag{20}$$

Thus, both candidate models included variance among dams and differed only in the presence/absence of sire effects.

We fit both candidate models for males and females using Stan[52] in R (ref. 53), and used the WAIC to determine the best-fitting model[54,55] (Supplementary Table 3 and Supplementary Fig. 3).

**Comparing pre- and post-invasion dispersal kernels.** We fit dispersal kernels to spread data from the first generation of our experimental invasions. The goal of this analysis was twofold. First, we tested whether there were any treatment-specific differences in dispersal at the beginning of the experiment; because replicate invasions were drawn from a common source population and randomly assigned to treatments, we expected control and shuffle dispersal kernels to be similar at the start of range expansion. Second, we compared the change in the mean dispersal distance from the beginning to the end of the experiment, which identifies the magnitude and direction of evolutionary change in dispersal ability at the level of individual invasion replicates.

At the start of the experiment, invasion replicates were initialized with five male and five female beetles in each of the first five patches of the dispersal array. Since beetles were simultaneously dispersing from multiple patches and not released from a single point, we could not estimate dispersal kernels using the same approach that we used in the common garden experiment. Instead, we estimated dispersal kernels from the first generation by fitting density data to mixture distributions composed of multiple dispersal kernels. To calculate these mixtures, we assumed that, within a replicate, beetles in each patch dispersed according to the same PIG dispersal kernel. We defined this kernel, $K$, as the probability mass function (pmf) of a 1D PIG distribution that is transformed to be symmetric about 0 and scaled so that it sums to 1:

$$K(x|\xi, \omega) = \frac{1}{2}\text{PIG}(|x||\xi, \omega) \tag{21}$$

where $K$ gives the probability that an individual travels $x$ patches, $\xi$ and $\omega$ are the mean and shape parameters of the PIG distribution, respectively, and 1/2 is a scaling factor to ensure that $K$ sums to 1.

Next, we defined the mixture distribution $M$ as a mixture of kernels $K$, each centred on a different starting patch and scaled so that $M$ sums to 1:

$$M(x|N, \xi, \omega) = \frac{1}{N}\sum_{i=0}^{N} K((x-i)|\xi, \omega) \tag{22}$$

where $N$ is the number of contiguous patches that beetles dispersed from (for our experimental design $N = 5$). Note that Equation 8 assumes that there are initially equal beetle densities in each starting patch, which was the case in our experiment. This mixture distribution enables us to use invasion-wave data (patch-specific densities following the first bout of dispersal, but prior to reproduction) to infer the dispersal kernel that generated the wave shape.

We tested whether this method could accurately estimate the PIG parameters $\xi$ and $\omega$ using simulated data for a region of parameter space appropriate for our system, and were able to reliably recover the parameters of the underlying kernels. We then fit dispersal data from the first generation of the experiment. We estimated male and female kernels separately, as we did when estimating dispersal kernels from the common garden experiments.

First, we tested for any treatment-specific differences in $\xi$, $\omega$ and random effect variances, just as we did when testing for dispersal differences in the common garden experiment. The models converged well and gave parameter estimates on scales comparable to parameter estimates from our common garden experiment. We fit candidate models using Stan[52] in R (ref. 53), and used WAIC to determine the best-fitting model[54,55] (Supplementary Table 6 and Supplementary Fig. 6).

Second, we compared the mean dispersal distance ($\xi$) between the first bout of dispersal (pre-invasion) and the first generation of the common garden experiment (post-invasion) for both treatments (Supplementary Fig. 8). For the 'post-invasion' estimates, we only considered the first common garden generation, as data from the second generation were qualitatively similar (Supplementary Tables 4 and 5; Supplementary Figs 4 and 5).

Although the results from these analyses are consistent with our other findings and theoretical expectations, limitations of this analysis warrant consideration. Notably, the dispersal conditions of the first generation were different from the conditions of the common garden experiment, and the first-generation kernels were estimated without population-level replication. While our multiple lines of evidence from disparate sources demonstrate that evolution resulted in divergent

dispersal ability, we suggest that readers interpret results from these first-generation dispersal kernel analyses with appropriate caution.

**Comparing invasion bottleneck size between treatments.** We measured population bottleneck sizes each generation, for each replicate, to determine whether the observed differences in invasion speeds could be due to treatment-specific differences in population bottlenecks. We estimated bottleneck size as the number of female beetles in the farthest patch of the expanding invasion wave, since only patches that contain females will make genetic contributions to future generations. We assumed that the number of leading-patch females was Poisson-distributed, and visual inspection confirmed that this choice of distribution was appropriate for the data. We performed model selection to determine whether treatment, generation or an interaction between treatment and generation were strong predictors of bottleneck size. We modelled bottleneck size $B_{jkl}$ for replicate $j$ undergoing treatment $k$ in generation $l$ as:

$$B_{jkl} \sim \text{Poisson}\left(\lambda_{jkl}\right) \tag{23}$$

$$\lambda_{jkl} = \beta_0 + \beta_{\text{TRT}}\text{TRT}_k + \beta_{\text{GEN}}\text{GEN}_l + \beta_{\text{TRT}\times\text{GEN}}\text{TRT}_k\text{GEN}_l + \gamma_j \tag{24}$$

$$\gamma_j \sim \text{Normal}\left(0, \sigma^2\right) \tag{25}$$

where $\lambda_{jkl}$ is the expected bottleneck size, $\beta_0$ is the model's intercept, $\beta_{\text{TRT}}$ represents the effect of the shuffle treatment, $\beta_{\text{GEN}}$ represents effect of generation and $\beta_{\text{TRT}\times\text{GEN}}$ represents the interaction between treatment and generation. A replicate-specific random effect in bottleneck size, $\gamma_j$, is centred at 0 and has variance $\sigma^2$. We fit all possible combinations of fixed effects in candidate models using the 'lme4' package[62] in R, and used AIC to select the best-fitting model (Supplementary Table 8).

**Data availability.** The data that support the findings of this study are available in the Dryad Digital Repository: http://dx.doi.org/10.5061/dryad.13410.

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

## Acknowledgements

Funding for this work was provided by NSF-DEB-1501814, NSF Data Analysis and Visualization Cyberinfrastructure grant OCI-0959097, and the Godwin Assistant Professorship at Rice University. We thank E. Castro, J. Cueller, M. Donald, A. Geiger, Z. Matranga, L. Monterroso, O. Nixon, R. Patterson and C. Thomas for their help in conducting the experiment. We also thank A. Bibian, A. Compagnoni, M.H. Downey, K. Ensor, R. Hufbauer, J. Levine, B. Melbourne, J. Saltz, E. Schultz, E. Siemann, M. Sneck, V. Rudolf, J. Strassman, T. Weiss-Lehman, J. Williams and D. Queller for comments on the project and manuscript.

## Author contributions

B.M.O. and T.E.X.M. contributed equally to experimental design, data analysis and writing the manuscript. B.M.O. conducted the experiment.

## Additional information

**Competing financial interests:** The authors declare no competing financial interests.

