## [Peer Review File · Nature Communications]

Reviewers' comments:

Reviewer #1 (Remarks to the Author):

This is one of two MSs I have been asked to review on empirical studies of the rapid evolution of accelerating rates of range expansion. In both of these manuscripts, studies on insects provide convincing evidence that evolutionary processes (of which spatial sorting is likely to be the most important) can magnify the variance and increase the mean rate of dispersal within a few generations. This is a robust result from mathematical analyses, and consistent with evidence from field invasions based on retrospective data, but not previously supported by direct experimental studies.

The work is solid, well-written and convincing. My personal opinion is that in an attempt to cover the myriad evolutionary phenomena that can influence evolutionary trajectories at an invasion front, the likely pre-eminence of the simplest (spatial sorting) is somewhat hidden because it is evaluated side-by-side with a host of other processes that are potential (but probably less important) contributors to the effects on dispersal rate. In their discussion of the impact of range-expansion on reproductive rates, the authors do not mention the following recent paper, which provides field data to show lower reproductive rates of range-edge individuals:

Hudson, C. H., B. L. Phillips, G. P. Brown, and R. Shine. 2015. Virgins in the vanguard: low reproductive frequency in invasion-front cane toads. *Biological Journal of the Linnean Society* 116:743-747.

Reviewer #2 (Remarks to the Author):

I reviewed both manuscripts by Weiss-Lehman et al. and Ochoki & Miller. The two studies are very similar in their aims and experimental manipulations using different beetle species and different statistical analyses. Generally, I think it is a clear strength that independent studies show very similar results and I would like to see more of those back-to-back publications. Both studies show interesting results that could make a significant contribution to the field of invasion biology.

As both studies used the same experimental design, both studies suffer however from the same problems when it comes to the interpretation of the data (as you will also see from the individual reports). Although the studies were set up as experimental evolution studies, there are no data supporting the rapid evolution of the traits as data - from controlled common garden experiments - are only presented from the end of the experiment. It is thus impossible to interpret whether and in which direction the populations in the two treatments evolved. If the authors actually collected these data but did not present these, these manuscripts could contribute significantly to the field. Without these data, I am afraid that a clear interpretation of the data is not possible and I would recommend to reject the manuscripts for publication in a *Nat Commun.* Both manuscripts also emphasize the loss of alleles to be important and to explain the higher variance in the evolved treatment but there is only very poor support from the data for the loss of alleles. This should be confirmed by additional data. Both manuscripts need also further clarification for a general audience.

Review: Rapid evolution of dispersal ability makes biological invasions faster and more variable

Ochocki and Miller present an interesting study on rapid evolution of dispersal using an experimental evolution approach with bean beetles. They compared dispersal speed and its variability of replicated populations that were either allowed to evolve over 10 generations or not. For the latter, they shuffled

individuals from all patches every generation and distributed these randomly to the patches keeping sex ratios and density similar to what they found before shuffling. Populations of the evolution treatment were not manipulated. The experiment confirms two main predictions from theory, that evolution matters as they found significant faster dispersal in the evolving populations (spatial-sorted) compared to the shuffled populations, and that dispersal is more variable in the evolving populations due to gene surfing. While I think this is a significant study with an interesting result, I have some reservations when it comes to the interpretation of the results given the presentation of the experimental design and results. Furthermore, the manuscript has several passages that need clarification.

Main points:

1. The power of experimental evolution stems from the fact that initially similar populations evolve differently under different treatments. Whereas the current study provides evidence for the differences between populations from the spatial sorting and shuffled treatment after 10 generations, there are no data provided for the initial replicated populations of generation 0. Are there actually differences in dispersal kernels and fecundity between the start and end of the experiment? Can you exclude that the shuffled population evolved to lower dispersal speed while the traits of the spatial-sorted populations stayed the same over the course of the experiment? The information of generation 0 is essential to interpret the results.
2. From the presentation of the experimental design and the results, it is not clear to me whether the result of the greater variability in dispersal speed is an artefact of the experimental design or not. The proposed mechanism of gene surfing depends on repeated bottlenecks with random loss of alleles. To what extent were the bottlenecks (i.e., the density of beetles in the leading edge patches) equal in the spatially sorted and shuffled treatment? Furthermore, can you exclude the possibility that the higher variance stems from a greater mean, as typically observed with count data (which you have here: number of patches dispersed).
3. My third main point is also related to the gene surfing mechanism. The argument for gene surfing is that alleles get randomly fixed at the leading edge even when they are deleterious. Dispersal is a 'complex, likely polygenic trait with imperfect heritability' (line 59) and the beetles are facultative sexually reproducing. You further also consider only patches with at least four beetles as the leading edge. How can alleles get fixed without strong selection for dispersal (homogenous patches, no novel environments)? There is also no data showing differences in alleles in the spatial-sorted populations.

Minor comments:

4. line 14: delete "spread"
5. line 16: delete "the"
6. line 22: use different word instead of "amplify"
7. line 23: explain "spatial evolutionary mechanism" better in the abstract
8. line 27: explain "dispersal behavior" better
9. lines 39-53: I think this paragraph needs clarification to make sure the reader follows the authors argumentation, e.g., the term 'spatial selection' is a general term in evolutionary biology and not only in the context of invasion biology. Furthermore, the difference between natural and spatial selection for higher growth rates is not clear. Selection for higher growth also requires resource competition.
10. Lines 58-60: the present study does not provide evidence that spatial selection varies with the level of heritability.
11. Lines 106-107: (and other places in the manuscript): it would be useful to explain that changes in the dispersal trait are stemming from genetic mixing and loss of alleles rather than novel mutations.
12. Lines 119-120: what do you mean by 'far exceeding the effect on the mean'. Why is this important?

13. Lines 185-188: are 7 beans limited food for the number of beetles found in this experiment? If not, it is not surprising that there is no evidence for changes in fecundity.
14. The leading edge was considered as patches with at least four beetles, whereas invasion success depends on the arrival of male and female at the leading edge. This needs better justification.
15. It is not clear why sometimes the 1st, sometimes also the 2nd generation of the common garden experiment is presented. How was the 2nd generation produced?

REVIEWERS' COMMENTS:

Reviewer #1 (Remarks to the Author):

The revisions have improved the paper; I think it is now excellent.

Here we summarize our major changes and responses, and below we elaborate on each of these points, including detailed responses to individual reviewer comments.

1. One reviewer argued that we cannot draw conclusions about the role of spatial selection without a contrast of trait values pre- and post-invasion. We respectfully disagree with this point for reasons that we elaborate below. We argue that our experimental design and data collection were the right approaches for our question, which focuses on the influence of spatial evolutionary processes on the ecological dynamics of spread and not on trait evolution *per se*.

2. While we are confident that the data presented in our first submission were sufficient to demonstrate the influence of spatial selection on spread dynamics, we now strengthen our conclusions with two new analyses requested by a reviewer. First, we show that there were no differences in dispersal kernels between treatments at the start of the experiment; together with the previously reported results of our common garden experiment, this result provides clear evidence that the differences observed at the end of the experiment are evolved differences. Second, we now provide a contrast of mean dispersal distance for each invasion replicate pre- and post-invasion, which enables us to document the magnitude and direction of evolutionary change for each replicate. These new results, presented in the Supplement, continue to support our original interpretation.

3. One reviewer made several additional requests for new analyses to address alternative hypotheses. We have added the new analyses that were requested.

Below, we structure our response to mirror the decision letter that we received. First, we respond to reviewer comments directed at the two papers, collectively. Accordingly, we and Weiss-Lehman et al. have prepared a shared response to these comments. Second, we (Ochocki and Miller) respond to the specific comments made about our manuscript. We reproduce the original decision letter in gray font and provide our responses in black font.

Comments applying to both manuscripts (joint responses in black font)

I reviewed both manuscripts by Weiss-Lehman et al. and Ochocki & Miller. The two studies are very similar in their aims and experimental manipulations using different beetle species and different statistical analyses. Generally, I think it is a clear strength that independent studies show very similar results and I would like to see more of those back-to-back publications. Both studies show interesting results that could make a significant contribution to the field of invasion biology.

As both studies used the same experimental design, both studies suffer however from the same problems when it comes to the interpretation of the data (as you will also see from the individual reports). Although the studies were set up as experimental evolution studies, there are no data

supporting the rapid evolution of the traits, as data - from controlled common garden experiments - are only presented from the end of the experiment. It is thus impossible to interpret whether and in which direction the populations in the two treatments evolved. If the authors actually collected these data but did not present these, these manuscripts could contribute significantly to the field. Without these data, I am afraid that a clear interpretation of the data is not possible and I would recommend to reject the manuscripts for publication in a Nat Commun..

This is an important point and we are grateful to the reviewer for raising it. We respectfully disagree with the reviewer's argument that "clear interpretation of the data" is not possible without trait data from the start of the experiment. There are essentially two types of contrasts that are relevant for studies of experimental evolution. One is the contrast of trait values before and after the operation of an evolutionary mechanism (e.g., selection), which is what the reviewer suggests is required. The second is the contrast between treatments in which an evolutionary mechanism does and does not operate, or in which different evolutionary mechanisms operate. The experiments in our two papers rely on the second type of contrast, following the precedent of previous high-profile studies of experimental evolution¹⁻⁷ and eco-evolutionary dynamics⁸⁻¹⁰. The experimental design and statistical procedures of both studies ensure that any changes in traits are due to the experimental treatments and the relative evolution of traits can be shown unambiguously. In particular, all of the replicates across all treatments were founded with random individuals from the same well-mixed population, which ensures that traits were the same on average at the beginning of the experiments. Indeed, the hypothesis tests for trait differences in the final generation quantify the reviewer's concern over whether evolution occurred, since they represent the probability of an effect as large or larger than the one observed, assuming the replicate populations were drawn at random from a larger well-mixed population. We suggest it is important to keep in mind the question that motivated these experiments: What is the influence of evolutionary mechanisms that arise from spatial genetic structure on the ecological dynamics of range expansion? This question demands an experimental design that facilitates contrast of ecological dynamics with and without the spatial sorting of alleles via dispersal, and this is precisely the contrast that our studies emphasize. We respectfully argue that our data allow us to answer this question rigorously and unambiguously. Many previous studies have demonstrated trait differences between range-core and range-edge populations consistent with theoretical predictions for spatial selection during range expansion¹¹⁻¹⁶; it was not our intent to recapitulate these results. What has remained unknown is the extent to which spatial evolutionary mechanisms modify the dynamics of expansion. This is precisely where our experiments make a new and important contribution. We have taken steps to clarify this point in both manuscripts.

The reviewer is correct that our studies were not designed to measure the absolute direction of evolution or to differentiate alternative hypotheses for the evolutionary trajectories of

demography and dispersal traits. We make this clearer in the individual manuscripts and exercise greater caution in our descriptions of evolved trait differences between the treatments, focusing on the relative differences between treatments post-invasion (specific changes cited below) and cautious discussion of the potential absolute direction of evolution.

Both manuscripts also emphasize the loss of alleles to be important and to explain the higher variance in the evolved treatment but there is only very poor support from the data for the loss of alleles. This should be confirmed by additional data.

As above, we respectfully argue that our studies were designed to isolate the effects of spatial genetic structure on the ecological dynamics of range expansion. The increases in variance detected by both studies were surprising results. We offer what we think is the most reasonable and well-supported interpretation of these results given existing theory and given our data: that the random loss or fixation of alleles at low-density expanding fronts ('gene surfing') amplified the variability across replicate invasions. In our revised manuscripts, we take greater care to emphasize that inferences about 'gene surfing' are interpretations of a surprising result, and not a mechanism that our studies were designed to test (specific changes cited below). We suggest that this would be an interesting and worthwhile direction for future study.

Both manuscripts need also further clarification for a general audience.

Both groups have clarified their papers for a general audience according to the reviewer's helpful suggestions (specific changes cited below).

Comments applying to Ochocki and Miller (responses in black font)

Response from the editor:

I sincerely apologize for the delay in the review process of your Nature Communications submission entitled "Rapid evolution of dispersal ability makes biological invasions faster and more variable." It has now been seen by 2 referees, whose comments are appended below. While they find your work of potential interest, they have raised substantive concerns that in our view need to be addressed before we can consider publication in Nature Communications.

Should further experimental data or analysis allow you to address these criticisms, we would be happy to look at a revised manuscript. However, I should stress that, because Nature Communications strives to provide an efficient editorial service and fast publication, we are reluctant to see manuscripts undergo multiple rounds of review. Please note that referee 2 would need to see stronger support for the main conclusion and interpretation on rapid evolution. As such, we would like to see the revised manuscript to address this and other concerns before we proceed to the next step.

Thank you for the opportunity to re-submit our manuscript according to your recommendations. As we describe below, we have responded to the reviewers' concerns by strengthening and clarifying our arguments, and by adding new data and analyses as requested. Specifically, we now provide stronger support for the interpretation of rapid evolution, including a new contrast of dispersal traits before and after range expansion for each invasion replicate. We believe these additions have strengthened our manuscript and satisfactorily address concerns raised by reviewers.

If the revision process takes significantly longer than three months, we will be happy to reconsider your paper at a later date, as long as nothing similar has been accepted for publication at Nature Communications or published elsewhere in the meantime.

When resubmitting your paper, we also ask that you ensure that your manuscript complies with our editorial policies.

Specifically, please ensure that the following requirements are met, and any relevant checklists are completed and uploaded with the revised article:

Reporting requirements for life sciences

research: http://www.nature.com/ncomms/authors/ncomms_lifesciences_checklist.pdf

We have also made changes to our manuscript to ensure that it complies with the editorial policies at Nature Communications, and have completed the relevant formatting checklists, which we have included with this submission.

Reviewer #1 (Remarks to the Author):

This is one of two MSs I have been asked to review on empirical studies of the rapid evolution of accelerating rates of range expansion. In both of these manuscripts, studies on insects provide convincing evidence that evolutionary processes (of which spatial sorting is likely to be the most important) can magnify the variance and increase the mean rate of dispersal within a few generations. This is a robust result from mathematical analyses, and consistent with evidence from field invasions based on retrospective data, but not previously supported by direct experimental studies.

The work is solid, well-written and convincing.

We thank the reviewer for this positive feedback.

My personal opinion is that in an attempt to cover the myriad evolutionary phenomena that can influence evolutionary trajectories at an invasion front, the likely pre-eminence of the simplest (spatial sorting) is somewhat hidden because it is evaluated side-by-side with a host of other processes that are potential (but probably less important) contributors to the effects on dispersal rate.

We have made modifications to the text so that the pre-eminent importance of spatial sorting is not lost in the discussion of other potential evolutionary processes (lines 52-55, 57-58, 94, 106, 16, and 217-220). However, we think it is important to acknowledge that other mechanisms of evolutionary change are also possible. We also think it is important to clarify the distinctions between "spatial sorting" and "spatial selection", which we continue to do (lines 47-50). This section may be a little tedious for an expert in this field, but we hope that clear definitions and comprehensive description of mechanisms will make the paper broadly accessible to non-specialist readers.

In their discussion of the impact of range-expansion on reproductive rates, the authors do not mention the following recent paper, which provides field data to show lower reproductive rates of range-edge individuals:

Hudson, C. H., B. L. Phillips, G. P. Brown, and R. Shine. 2015. Virgins in the vanguard: low reproductive frequency in invasion-front cane toads. *Biological Journal of the Linnean Society* 116:743-747.

We have now included references to the Hudson et al. (2015) paper (line 60). In addition, we have added a paragraph to the Discussion section (lines 217-239) which addresses how our results relate to those of Hudson et al. (2015), as well as other recent discoveries in this field. Thank you for this suggestion.

Reviewer #2 (Remarks to the Author):

Ochocki and Miller present an interesting study on rapid evolution of dispersal using an experimental evolution approach with bean beetles. They compared dispersal speed and its variability of replicated populations that were either allowed to evolve over 10 generations or not. For the latter, they shuffled individuals from all patches every generation and distributed these randomly to the patches keeping sex ratios and density similar to what they found before shuffling. Populations of the evolution treatment were not manipulated. The experiment confirms two main predictions from theory, that evolution matters as they found significant faster dispersal in the evolving populations (spatial-sorted) compared to the shuffled populations, and that dispersal is more variable in the evolving populations due to gene surfing. While I think this is a significant study with an interesting result, I have some reservations when it comes to the interpretation of the results given the presentation of the experimental design and results. Furthermore, the manuscript has several passages that need clarification.

Main points:

1. The power of experimental evolution stems from the fact that initially similar populations evolve differently under different treatments. Whereas the current study provides evidence for the differences between populations from the spatial sorting and shuffled treatment after 10 generations, there are no data provided for the initial replicated populations of generation 0. Are there actually differences in dispersal kernels and fecundity between the start and end of the experiment? Can you exclude that the shuffled population evolved to lower dispersal speed while the traits of the spatial-sorted populations stayed the same over the course of the experiment? The information of generation 0 is essential to interpret the results.

We understand this concern and thank the reviewer for their suggestion to include analyses that account for trait values in the first generation of invasion, which we think have improved our manuscript. In response to the reviewer's concerns, we have added new analyses to determine (1) whether there were differences in control and shuffle dispersal kernels at the start of the experiment, and (2) the magnitude and direction of evolutionary change in dispersal from the beginning to the end of the experiment for each replicate invasion. We focus on the dispersal kernel since this was the trait that responded to our manipulation of spatial sorting. Our results indicate that (1) there were no initial differences in dispersal ability between replicates assigned to control and shuffle treatments, suggesting that differences in dispersal at the end of the experiment were not simply the result of an

undetected sampling effect (lines 148-156 and Supplement lines 223-266; Supplement Figure 6, Supplementary Table 6); and (2) the magnitude and direction of evolutionary change in dispersal kernels is consistent with the results previously presented in the manuscript (Supplement lines 268-272; Supplementary Figure 8). Specifically, we found that some control replicates evolved greatly increased dispersal ability while others evolved decreased dispersal (Supplementary Figure 8), consistent with our original finding that spatial genetic structure leads to an increase in expansion speed, on average, and also an increase in variance across replicates.

We estimated the pre- and post-invasion dispersal kernels using slightly different methodologies (pre-invasion: Supplement lines 223-266; post-invasion: lines 377-386 and 396-429) and, accordingly, suggest that readers compare these kernels with appropriate caution (Supplement lines 274-280). For this reason and the reasons outlined in our joint response, we continue to emphasize the relative difference between treatments as the key contrast that supports our conclusions.

2. From the presentation of the experimental design and the results, it is not clear to me whether the result of the greater variability in dispersal speed is an artefact of the experimental design or not. The proposed mechanism of gene surfing depends on repeated bottlenecks with random loss of alleles. To what extent were the bottlenecks (i.e., the density of beetles in the leading edge patches) equal in the spatially sorted and shuffled treatment? Furthermore, can you exclude the possibility that the higher variance stems from a greater mean, as typically observed with count data (which you have here: number of patches dispersed).

We have added several new analyses to the manuscript to address these concerns. First, we conducted a new statistical analysis to determine whether densities at leading-edge patches were similar between our two treatments (Supplement lines 282-301). We found that they were (lines 200-203; Supplementary Table 8). It is therefore unlikely that the greater variability of spatially sorted invasions was an artefact of our experimental design. However, it is important to note that the shuffle treatment experienced no serial genetic bottlenecks at the invasion fronts because a random set of alleles from the range core arrived at the invasion fronts in each generation. This is central to our interpretation, since the persistent randomization of alleles at the invasion fronts is expected to disrupt the 'gene surfing' mechanism, which relies on serial bottlenecks. We make this logic clearer at lines 200-203.

Second, to address the potential relationship between mean and variance of spatial extent, we have added an analysis of the coefficient of variation (CV) in invasion extent through time (lines 356-360). The results show that spatially sorted invasions had greater CV in invasion extent than shuffled invasions, suggesting that the increase in variance of sorted invasions is unlikely to be driven solely by the increase in the mean (lines 117-119, Supplementary

Figure 3). Furthermore, we emphasize that our original linear mixed model-based analyses are focused on invasion velocity, not spatial extent, and these analyses are designed to isolate treatment effects on variance independent of the mean (lines 306-357).

3. My third main point is also related to the gene surfing mechanism. The argument for gene surfing is that alleles get randomly fixed at the leading edge even when they are deleterious. Dispersal is a 'complex, likely polygenic trait with imperfect heritability' (line 59) and the beetles are facultative sexually reproducing. You further also consider only patches with at least four beetles as the leading edge. How can alleles get fixed without strong selection for dispersal (homogenous patches, no novel environments)? There is also no data showing differences in alleles in the spatial-sorted populations.

First, there seems to be a misunderstanding of our experimental design, which we have taken steps to clarify in our revision. We identified the invasion front as the most distant patch with four or more beetles solely for the purposes of statistical analysis: analysis of invasion velocity demanded a standardized way to measure displacement of the invasion front. As we describe in the text (lines 310-318), the choice of four beetles was arbitrary, other density thresholds yielded similar results, and this approach is consistent with similar analyses in the literature^{17,18}. With a threshold of four, there were almost always beetles ahead of the "invasion front" (as we define it) that contributed to the dynamics we observed. Indeed, it is these very low density patches (often one or two individuals) where we expect genetic bottlenecks were strongest. We have clarified these points in the manuscript (lines 196-203).

Second, the reviewer wonders how alleles can become fixed without selection. This is the essence of the "gene surfing" hypothesis, which we clearly did not describe effectively in our first submission. We emphasize that this is not a new idea and it is not our idea. The population genetics literature provides a strong body of theory for 'gene surfing' (cited in our manuscript), and we use this theory to interpret the evolved increase in variance across invasion replicates. Because expansion events are driven by relatively few long-distance colonists, leading-edge patches of expanding populations are subject to strong genetic bottlenecks. Iterating this process through time, the leading edge colonists of a given generation are likely to be offspring from the previous generation's leading edge, thus propagating the genetic bottleneck. Importantly, this can lead to evolutionary changes at the leading edge via loss or fixation of alleles, even in a homogenous environment and even in the absence of selection; it is essentially genetic drift, in space. We have clarified this idea in several places (lines 191-215).

Finally, the reviewer is correct that we do not show data on allele frequencies in support of the gene surfing hypothesis. As described above in our joint response, we now emphasize that we invoke the gene surfing hypothesis as a potential mechanism to explain a surprising

result: the greater variability of spatially sorted invasions relative to shuffled invasions. Our study was not designed to test this mechanism, and it is possible that other or additional mechanisms contributed to the result. We make this clearer in the text (line 190-202).

Minor comments:

4. line 14: delete "spread"

We have made this change, although after formatting the abstract to meet *Nature Communications* requirements, this sentence is no longer in the abstract and appears in the opening paragraph of the introduction (line 28).

5. line 16: delete "the"

After formatting the abstract to meet *Nature Communications* requirements, we have removed this sentence, because this sentence appears in the second paragraph of the introduction (lines 38-40):

“Recent theory suggests that evolutionary processes unique to spreading populations can influence the ecological dynamics of invasion by modifying traits related to dispersal, reproduction, or both.”

6. line 22: use different word instead of "amplify"

In formatting the abstract to meet *Nature Communications* requirements, we rewrote this sentence so that “increased” now refers to both invasion speed and invasion variability (lines 19-23):

“We show that spatial sorting promotes rapid evolution of dispersal distance, which increases the speed and variability of replicated invasions...”

7. line 23: explain "spatial evolutionary mechanism" better in the abstract

Instead of referring to “spatial evolutionary mechanisms” in the abstract, we now refer to “spatial sorting”, which identifies the specific manipulation that we imposed. We also define spatial sorting more explicitly in the abstract (lines 15-17).

8. line 27: explain "dispersal behavior" better

We have replaced “dispersal behavior” with “dispersal distance”, which more explicitly describes the result that we are reporting (lines 23-24).

9. lines 39-53: I think this paragraph needs clarification to make sure the reader follows the authors argumentation, e.g., the term 'spatial selection' is a general term in evolutionary biology and not only in the context of invasion biology. Furthermore, the difference between natural and spatial selection for higher growth rates is not clear. Selection for higher growth also requires resource competition.

We thank the reviewer for their suggestion to clarify the use of “spatial selection”. We have worked to clarify that our use of "spatial selection" is consistent with the way the process has been previously described in the literature (lines 45-50). We also clarify the distinction between natural and spatial selection on growth rate (lines 50-52).

10. Lines 58-60: the present study does not provide evidence that spatial selection varies with the level of heritability.

It is true that we do not provide evidence that responses to spatial selection vary with the level of heritability. Our point is simply that response to a selective force will be weakened by imperfect heritability. Given that most theory for spatial selection has assumed perfect heritability, it is relevant to acknowledge that empirical estimates for heritability of dispersal are modest (with h^2 estimates on the range of 0.28 to 0.61)^{11,19-21}, and thus the effects of spatial selection in real systems may be weaker than predicted by theory. This is a key motivation for our study. We clarify these points in our revised manuscript (lines 63-67) and add additional citations to document empirical estimates of dispersal heritability.

11. Lines 106-107: (and other places in the manuscript): it would be useful to explain that changes in the dispersal trait are stemming from genetic mixing and loss of alleles rather than novel mutations.

While changes in the dispersal trait can be caused by novel mutations, the reviewer makes a good point that most of the changes observed in our system are likely to be due to genetic mixing and change in frequency of existing alleles across space. We have made this clarification more apparent in lines 186-189.

12. Lines 119-120: what do you mean by 'far exceeding the effect on the mean'. Why is this important?

Our intention was simply to highlight that the proportional change in among-replicate variance was much larger than the proportional change in mean invasion speed – a surprising result given that an increase in invasion speed is the main consequence of spatial sorting predicted in the theoretical literature. We have added text to elaborate on the general

importance of this finding, and the likely cause of it (gene surfing) in the discussion (lines 191-239) We have also rewritten the sentence to improve its clarity (lines 172-174).

13. Lines 185-188: are 7 beans limited food for the number of beetles found in this experiment? If not, it is not surprising that there is no evidence for changes in fecundity.

When more than one female beetle is present, an environment containing seven beans is resource-limited. We expect intraspecific competitive ability to be important in these environments and to result in density-dependent population growth. Our data show that, in a seven-bean resource environment, these populations have a carrying capacity of roughly 40 beetles (Figure 1), and we have highlighted that in the text (lines 280-282 and 301-304). Thus, beetles in low-density patches at the leading edge experience a release from density dependence, and theory suggests that these individuals may experience selection for increased fecundity. We have clarified this description in the text (lines 280-284).

14. The leading edge was considered as patches with at least four beetles, whereas invasion success depends on the arrival of male and female at the leading edge. This needs better justification.

As described above (point #3), we designated “leading patches” as the farthest patch that contained at least four beetles simply for the purpose of estimating invasion wave speed. The reviewer is correct that success of leading-edge patches requires both sexes, and our previous work on sex structure during range expansion shows that mating failure at the leading edge is an important element of beetle invasions¹⁷. Our approach in the present study implicitly accounts for mating failure: since unmated beetles produce no offspring to disperse, mating failure modifies the wave shape in the next generation, and hence the invasion speed. Our analysis makes no assumption that the designated “invasion front” will be viable; it is simply a place-marker on the wave front. We would have arrived at qualitatively similar results had we chosen any density below the carrying capacity, even if this density was represented by, say, a solitary male. We have clarified this at lines 310-318.

15. It is not clear why sometimes the 1st, sometimes also the 2nd generation of the common garden experiment is presented. How was the 2nd generation produced?

We are very grateful to the reviewer for identifying this issue with our discussion of the common garden results. The results from both common garden generations were qualitatively similar, so we chose to present only results from the first common garden generation in the main text to simplify the presentation and interpretation for a general audience. We have clarified this in the text (lines 140-146, 160-161, Figures 3 and 4). Our Supplementary Information, on the other hand, provides full details of both common garden generations.

To produce the second common garden generation, we randomly selected 10 males and 10 females from the first common garden generation and transferred them to an environment with conditions identical to the first generation. This has been clarified in the text (lines 371-374).

References

1. Huey, R. B., Partridge, L. & Fowler, K. Thermal sensitivity of *Drosophila melanogaster* responds rapidly to laboratory natural selection. *Evolution* **45**, 751 (1991).
2. Gilchrist, G. W., Huey, R. B. & Partridge, L. Thermal sensitivity of *Drosophila melanogaster*: evolutionary responses of adults and eggs to laboratory natural selection at different temperatures. *Physiol. Zool.* **70**, 403–414 (1997).
3. Collins, S. & Bell, G. Phenotypic consequences of 1,000 generations of selection at elevated CO₂ in a green alga. *Nature* **431**, 566–569 (2004).
4. Burke, M. K. *et al.* Genome-wide analysis of a long-term evolution experiment with *Drosophila*. *Nature* **467**, 587–590 (2010).
5. Tomkins, J. L., Hazel, W. N., Penrose, M. A., Radwan, J. W. & LeBas, N. R. Habitat complexity drives experimental evolution of a conditionally expressed secondary sexual trait. *Curr. Biol.* **21**, 569–573 (2011).
6. Egan, S. P. *et al.* Experimental evidence of genome-wide impact of ecological selection during early stages of speciation-with-gene-flow. *Ecol. Lett.* **18**, 817–825 (2015).
7. Huang, Y., Stinchcombe, J. R. & Agrawal, A. F. Quantitative genetic variance in experimental fly populations evolving with or without environmental heterogeneity. *Evolution* **69**, 2735–2746 (2015).
8. Yoshida, T., Jones, L. E., Ellner, S. P., Fussmann, G. F. & Hairston, N. G. Rapid evolution drives ecological dynamics in a predator–prey system. *Nature* **424**, 303–306 (2003).
9. Duffy, M. A. & Sivars-Becker, L. Rapid evolution and ecological host-parasite dynamics. *Ecol. Lett.* **10**, 44–53 (2007).
10. Palkovacs, E. P. *et al.* Experimental evaluation of evolution and coevolution as agents of ecosystem change in Trinidadian streams. *Philos. Trans. R. Soc. B Biol. Sci.* **364**, 1617–1628 (2009).
11. Phillips, B. L., Brown, G. P. & Shine, R. Evolutionarily accelerated invasions: the rate of dispersal evolves upwards during the range advance of cane toads. *J. Evol. Biol.* **23**, 2595–2601 (2010).

12. Phillips, B. L., Brown, G. P., Webb, J. K. & Shine, R. Invasion and the evolution of speed in toads. *Nature* **439**, 803–803 (2006).
13. Lindström, T., Brown, G. P., Sisson, S. A., Phillips, B. L. & Shine, R. Rapid shifts in dispersal behavior on an expanding range edge. *Proc. Natl. Acad. Sci.* **110**, 13452–13456 (2013).
14. Fronhofer, E. A. & Altermatt, F. Eco-evolutionary feedbacks during experimental range expansions. *Nat. Commun.* **6**, 6844 (2015).
15. Therry, L., Nilsson-Örtman, V., Bonte, D. & Stoks, R. Rapid evolution of larval life history, adult immune function and flight muscles in a poleward-moving damselfly. *J. Evol. Biol.* **27**, 141–152 (2014).
16. Lombaert, E. *et al.* Rapid increase in dispersal during range expansion in the invasive ladybird *Harmonia axyridis*. *J. Evol. Biol.* **27**, 508–517 (2014).
17. Miller, T. E. X. & Inouye, B. D. Sex and stochasticity affect range expansion of experimental invasions. *Ecol. Lett.* **16**, 354–361 (2013).
18. Wagner, N. K., Ochocki, B. M., Crawford, K. M., Compagnoni, A. & Miller, T. E. X. Genetic mixture of multiple source populations accelerates invasive range expansion. *J. Anim. Ecol.* (2016). doi:10.1111/1365-2656.12567
19. Hansson, B., Bensch, S. & Hasselquist, D. Heritability of dispersal in the great reed warbler. *Ecol. Lett.* **6**, 290–294 (2003).
20. Saastamoinen, M. Heritability of dispersal rate and other life history traits in the Glanville fritillary butterfly. *Heredity* **100**, 39–46 (2007).
21. Doligez, B., Gustafsson, L. & Part, T. ‘Heritability’ of dispersal propensity in a patchy population. *Proc. R. Soc. B Biol. Sci.* **276**, 2829–2836 (2009).

REVIEWER'S COMMENTS: Reviewer #1 (Remarks to the Author): The revisions have improved the paper; I think it is now excellent.

We thank the reviewer for this positive feedback.